# Association of hypoglycemic events with cognitive impairment in patients with type 2 diabetes mellitus: Protocol for a dose-response meta-analysis

**Min Ye**[1], **Ai Hong Yuan**[2]*, **Qi Qi Yang**[1], **Qun Wei Li**[1], **Fei Yue Li**[1], **Yan Wei**[1]

**1** The First School of Clinical Medicine, Anhui University of Chinese Medicine, Hefei, Anhui, China,
**2** Acupuncture and Rehabilitation Department, The First Affiliated Hospital of Anhui University of Chinese Medicine, Hefei, Anhui, China

* 490603279@qq.com, yuanah2023@163.com

## Abstract

### Introduction

With an incidence rate as high as 46%-58%, hypoglycemia is a common complication of glycemic management among those suffering from type 2 diabetes mellitus(T2DM). According to preclinical research, hypoglycemia episodes may impair cognition by harming neurons. However, there is still controversy regarding the clinical evidence for the relationship between hypoglycemic events and the likelihood of cognitive impairment. Furthermore, little research has been done on the dose-response association between hypoglycemia incidents and the possibility of cognitive impairment. To address these knowledge gaps, the present research intends to update the comprehension of the association among hypoglycemic events and the risk of cognitive impairment and to clarify the correlation between dose and response by incorporating the most recent investigations.

### Method and analysis

This work has developed a protocol for a systematic review and meta-analysis that will examine, via a well-organized assessment of several databases, the relationship between the incidence of hypoglycemia and the probability of cognitive impairment. Observational studies investigating the connection between hypoglycemia episodes and cognitive impairment will be included. The databases that will be searched are PubMed, Web of Science, the Chinese Biomedical Literature Database (CBM), Cochrane Library, Embase, the China National Knowledge (CNKI), Wan Fang, the Chinese Science and Technology Periodical Database (VIP), and Du Xiu. Literature from the establishment of each database to December 2023 will be included in the search. Two researchers will independently screen the studies that satisfy the requirements for both inclusion and exclusion. A third researcher will be asked to mediate any disputes. The methodological caliber of the studies included will be assessed utilizing the Newcastle-Ottawa Scale (NOS) or the Joanna Briggs Institute (JBI) critical appraisal method. With regard to GRADE, which stands for Grading of

**Data Availability Statement:** Our paper belongs to the meta-analysis protocol, and this paper itself does not involve any data at present. In the process

of formulating the protocol, we have pre-searched relevant databases and extracted partial data from relevant literature to ensure the feasibility of the research protocol. Since the relevant data have not been uploaded to public repositories, we will upload the pre-searched relevant data as Supporting Information files.

**Funding:** This work was supported by the Natural Science Foundation of Anhui Province (Grant No.2108085MH308), the National Inheritance Studio of Famous and Veteran TCM Experts (Grant No. 8187151181) and the Fifth Batch of the National TCM Clinical Excellent Talents Training Program. AHY received funding from the above projects. The funders did not and will not have a role in study design, data collection and analysis, decision to publish, or preparation of the manuscript.

**Competing interests:** The authors have declared that no competing interests exist.

**Abbreviations:** T2DM, Type 2 diabetes mellitus; GRADE, Grading of Recommendations, Assessment, Development, and Evaluation; CBM, the Chinese Biomedical Literature Database; ROBIS, Risk of Bias in Systematic Reviews; CNKI, the China National Knowledge; VIP, the Chinese Science and Technology Periodical Database; NOS, the Newcastle-Ottawa Scale; JBI, the Joanna Briggs Institute; PROSPRO, Prospective register of systematic reviews; BMI, Body Mass Index; MOOSE, Meta-analyses of Observational Studies in Epidemiology; PRISMA-P, the Preferred Reporting Items for Systematic Reviews and Meta-Analysis Protocols statement; OR, odds ratio; RR, risk ratio; HR, hazard ratio; CI, confidence intervals; MeSH, Medical Subject Headings.

Recommendations, Assessment, Development, and Evaluation, the quality of the evidence will be evaluated. ROBIS Tool will be used to evaluate the risk of bias in the development of the systematic review. If the data is accessible, meta-analysis and dose-response curve analysis will be employed by Stata software. However, if the data does not allow for such analysis, a descriptive review will be performed.

## Discussion and conclusion

Hypoglycemic episodes may raise the likelihood of cognitive impairment, according to earlier investigations. This study will update the relevant evidence and explore the dose-response connection between hypoglycemic episodes and cognitive impairment. The results of this review will have significant effects on decision-making by individuals with diabetes, healthcare providers, and government policy institutions.

## Trial registration

**Prospero registration number:** CRD42023432352.

## Introduction

Type 2 diabetes mellitus (T2DM) is a chronic metabolic condition marked by insulin resistance and enhancing loss of β-cell function [1]. It is one of the important subtypes of diabetes, accounting for more than 90% of the diabetic population. T2DM has a high incidence, numerous complications, and imposes a heavy medical burden [2–4]. The increasing number of people with T2DM has become a great pressure and challenge in the field of public health due to the development of an ageing society [5].

Cognitive dysfunction is one of the most common complications of T2DM [6]. Empirical research has shown that individuals experiencing T2DM have a 2–3 times higher risk of developing cognitive disorders such as vascular dementia and Alzheimer's disease compared to healthy individuals [7, 8]. The overall incidence of mild cognitive dysfunction in T2DM patients is as high as 45% [9]. However, currently, there is no officially recognized effective treatment for cognitive impairment caused by T2DM worldwide. Investigating practical preventative measures for cognitive dysfunction caused by T2DM will therefore be particularly crucial.

Blood glucose control is the core goal of the clinical therapy strategy for individuals concerning T2DM, and iatrogenic hypoglycemia is a common complication in the process of blood glucose control in patients with T2DM, with an incidence rate as high as 46%-58% [10]. The immediate effect of hypoglycemia on cognitive ability is well known to be very significant, with typical manifestations including sluggish response, lethargy, coma, etc [11]. However, the long-term impairment and cumulative effect of hypoglycemic events on cognitive function, as well as the dose-response relationship, remain unknown. In animal models, it has been observed that recurrent hypoglycemic events seem to cause cumulative brain damage, which is strongly connected to the eventual development of chronic cognitive dysfunction [12–14]. In recent years, clinical investigations have been carried out to examine the connection between hypoglycemic events and subsequent cognitive impairment, but unfortunately, different investigations have shown almost opposite conclusions. Studies such as Wajd Alkabbani et al. [15–18] have suggested that hypoglycemia significantly increases the risk of cognitive impairment,

whereas Tali Cukierman-Yaffe et al. [19–21] found that hypoglycemia does not impair cognitive function or increase the likelihood of cognitive dysfunction. Therefore, the connection between hypoglycemic events and cognitive impairment remains controversial.

In 2022, Maria Dolores Gomez-Guijarro's team [22] executed a systematic review and meta-analysis of research publications on the connection between severe hypoglycemic incidents and dementia in individuals concerning T2DM. However, due to the lack of relevant research at the time and the high heterogeneity among the literature, they were only able to incorporate 7 pieces of suitable literature in the evaluation. Simultaneously, a substantial portion of the literature indicates a moderate susceptibility to bias, as acknowledged by the authors who emphasized the need for cautious interpretation of their findings. Consequently, there is an absence of dependable evidence-based outcomes to substantiate the impact of hypoglycemic events on cognitive competence in individuals concerning T2DM. This manuscript aims to establish a methodological structure (protocol) to explore the cumulative effect of frequency of hypoglycemic events on cognitive function and further explore the dose-response relationship through systematic review and meta-analysis, which aims to update the association between hypoglycemia and cognitive impairment by including the latest relevant studies. Through these objectives, this study will provide new insights and strong evidence for the development of clinical glycemic control programs and the improvement of cognitive impairment prevention strategies.

## Methods and materials

Before undertaking this research, we duly registered the study protocol with the PROSPERO database (Registration number: CRD42023432352). The execution of this study will adhere to the procedure outlined in the Cochrane Handbook for Systematic Reviews of Interventions [23]. The reporting of this study will adhere to the guideline of Meta-analyses of Observational Studies in Epidemiology (MOOSE) [24], and the Preferred Reporting Items for Systematic Reviews and Meta-Analysis Protocols statement (PRISMA-P) will be applied for all subsequent reporting [25]. The PRISMA-P checklist is shown in the S1 Appendix.

### Search strategy

A thorough search will be undertaken across various databases including PubMed, Web of Science, CBM, Cochrane Library, Embase, CNKI, Wan Fang, Wei Pu, and Du Xiu, covering the period from inception to December 2023. The search will utilize Medical Subject Headings (MeSH) and free terms, such as 'diabetes mellitus, type 2', 'noninsulin', 'noninsulin-dependent diabetes mellitus', 'type 2 diabetes', 'type 2 diabetes mellitus', 'type 2 diabetic', 'T2DM', 'DM', 'cognitive dysfunction', 'cognition disorders', 'cognitive disorder', 'dementia', 'cognitive decline', 'cognition disorder', 'cognitive deficit', 'cognitive impairment', 'executive function', 'cognitive function', 'memory', 'neurodegeneration', 'neurodegenerative disease', 'neurocognitive disorder', 'neuropsychiatric disorder', 'mental deterioration', 'dysmnesia', 'learning-memorizing ability', 'allomnesia', 'risk factor', 'predicted', 'predictor', 'risk', 'relat', 'associat', 'factor', 'reason', 'correlated', 'predictor', 'influen', 'inciden', 'relevan', for comprehensive coverage. Furthermore, reference lists of relevant papers will be manually checked and records from relevant trial registries will be retrieved. The search strategy will finally be formulated based on the results of repeated pre-search to include relevant studies as comprehensively as possible. The search strategy is shown in S2 Appendix.

### Eligibility criteria

**Inclusion criteria.** Studies meeting all the given requirements will be contained.

**Type of studies.**   All observational research will be included, including cross-sectional studies, case-control studies, and cohort studies.

**Type of participants.**   Individuals who have been medically diagnosed with type 2 diabetes and have been at least 18 years old are eligible to participate. Prior to enrollment and at the time of enrollment, every participant had normal cognitive function. The study has no restrictions on sex, race, duration of diabetes, or severity of diabetes among the participants.

**Criteria for the assessment of hypoglycemia.**   The plasma glucose level was less than 3.9 mmol per liter [11], the patient showed symptoms such as palpitation and dizziness, and the diagnosis was established by a trained physician or a formal medical institution.

**Criteria for the assessment of cognitive impairment.**   The International Classification of Diseases (ICD) of the World Health Organization, the Diagnostic and Statistical Manual of Mental Disorders (DSM) of the American Psychiatric Association, cognitive testing, or other nationally recognized diagnostic standards were utilized to make diagnoses [26–28]. A qualified medical professional or other recognized healthcare provider was responsible for the diagnosis. The sources of diagnoses were either medical claims archives or electronic case databases.

**Contents of studies.**   Studies that explored the connection between hypoglycemia and the likelihood of cognitive dysfunction will be included. The included studies have to provide data on the connection between the two, such as odds ratio (OR), risk ratio (RR), hazard ratio (HR), and 95% confidence intervals (CI).

**Exclusion criteria.**   Studies will be disqualified if they fulfill any of the specified criteria.

1. Duplicate publication of the same study (studies that have more detailed and credible results, are more recent, or have a larger sample size will be selected);

2. Studies for which full text or relevant data are not available;

3. Studies with insufficient methodological detail or poor quality (e.g., high risk of bias, inadequate statistical analysis);

4. Studies with comorbid conditions that could independently affect cognitive function (e.g., neurodegenerative diseases other than diabetes-related cognitive impairment);

5. Studies where the dose or severity of hypoglycemic events is not clearly defined;

6. Studies not published in English or Chinese;

7. Studies with a duration that is too short to capture meaningful cognitive changes;

8. Studies with significant heterogeneity in interventions or treatment protocols that may impact the ability to conduct a meaningful dose-response analysis.

## Literature screening

The literature will be managed using EndNote software. Two reviewers will screen the literature independently. Duplicate literature will be eliminated first. Then, the title and abstract will be used to weed out the literature that does not support the research theme. After that, the other literature will next be carefully read in its entirety and evaluated in light of the inclusion and exclusion criteria. If there is a difference of opinion, a compromise will be decided after chatting with another reviewer. To present the findings of the research screening and selection procedure, we will use the PRISMA flowchart (S1 Fig).

## Data extraction

The information that followed will be obtained independently by two researchers.

1. General information of studies: the first author, year of publication, country and region, study type, and duration of follow-up;

2. Participant characteristics: average age, physical activity levels, body mass index (BMI), sample size;

3. Disease characteristics: diagnostic criteria for hypoglycemia, duration of hypoglycemia, degree of hypoglycemia (light, resolves with rest; medium, resolves with medicine; severe, resolves after receiving comprehensive hospital treatment), measurements of hypoglycemia, frequency of hypoglycemic events, duration of diabetes mellitus, measurements of diabetes mellitus, definition of cognitive impairment, measurements of cognitive impairment, types of cognitive impairment(it may consist of Alzheimer's disease, vascular dementia, and other types.), and diagnostic criteria for cognitive impairment.

## Evaluation of literature quality

The Newcastle-Ottawa Scale (NOS) and the Joanna Briggs Institute (JBI) critical assessment method will be used by two investigators to independently grade the methodological quality of the included studies. Case-control study's primary evaluation criteria focus on subject selection (4 points), group comparability (2 points), and exposure factor measurement (3 points). Cohort study grading criteria include subject selection (4 points), group comparability (2 points), and outcome measurement (3 points). The quality of studies will be judged and categorized as 'High', 'Medium', or 'Low' based on the scoring criteria. The total score for both evaluations is 9 points. Scores above six indicate high quality, scores of five indicate medium quality, and scores below five indicate low quality. Eight items are used in the JBI critical assessment method to evaluate the cross-sectional survey. The responses to each item include four answers: "yes", "no", "not clear" and "not applicable". If a cross-sectional study has a "yes" response of more than 70%, the study is considered high-quality literature. A response of 50% to 69% is considered moderate quality, while less than 50% is considered low-quality literature.

## Data synthesis

If there are enough studies (at least 2) that define the exposure and the desired outcome similarly, meta-analysis will be carried out; if not, only descriptive analyses will be carried out. To conduct our statistical analysis, we will first use the DerSimonian-Laird method to generate the pooled RR and its 95% CI [29]. In cases where the literature does not provide the RR, we will employ specific formulas to convert OR and HR to RR [30–32]. To assess the heterogeneity of the literature, we will use the Chi-square test and $I^2$ statistic. There will be four categories for heterogeneity: not important (0%-40%), moderate (30%-60%), substantial (50%-90%) and considerable (75%-100%). The fixed effect model will be applied for combined analysis if the $I^2$ value is less than 50%. A random effect model will be applied when $P < 0.1$ or $I^2 > 50\%$ [33, 34]. Regardless of whether statistically significant heterogeneity is found, if there is enough literature, we will investigate clinical heterogeneity by subgroup analysis and other workable analytical techniques. For the statistical analysis, we will use the STATA SE program, version 15 (StataCorp).

## Subgroup analysis and meta-regression

We will conduct subgroup analysis to explore the sources of heterogeneity if there are enough subgroup studies [34]. Factors used for grouping may include age, sex, BMI, physical activity levels, duration of diabetes, the severity of diabetes, follow-up period, type of study, country of study, type of cognitive impairment, severity of hypoglycemia, frequency of hypoglycemia,

and duration of hypoglycemia. Furthermore, we will perform meta-regression to look at how these factors affect the study's final combined results when the number of relevant studies is more than ten.

### Sensitivity analysis

The stability of the combined findings will be identified by carrying out the sensitivity analysis, in which a single study will be removed one at a time.

### Dose-response curve analysis

If data are available, we will use dose-effect curve analyses to investigate the dose-response relationship between hypoglycemic events and the likelihood of cognitive impairment. The frequency of hypoglycemia, the severity of hypoglycemia, and the duration of hypoglycemia are examples of exposure variables, while the incidence of cognitive impairment is an example of an outcome measure.

### Publication bias

The retrieval of literature may generally involve omittance, small sample studies, published bias, negative results, etc., depending on the research purpose. Any of these conditions may contribute to publication bias, which will have an impact on the accuracy of the meta-analysis results. Therefore, to minimize publication bias as much as feasible, we will not only develop a strict search strategy but also gather pertinent grey literature as comprehensively as possible. Moreover, publication bias will be investigated using Begg's test and Egger's test. When there are more than ten papers in the meta-analysis, the evaluation will be presented in the form of funnel plots. A P value of less than 0.05 will be considered to indicate publication bias [35].

### Grading the quality of evidence

Using the GRADE approach, we will evaluate the strength of the evidence. Depending on the study design, bias risk, inconsistency, indirect evidence, imprecision, and publication bias, each outcome may receive a high, moderate, low, or extremely low evidence rating [36]. Evidence at a high level will result in "strong recommendation", evidence at a moderate level in "practice consideration", and evidence below a moderate level in "insufficient evidence to guide" [37]. One tool that is particularly suited for evaluating the risk of bias in systematic reviews is ROBIS. It can be used to evaluate the relationship between the practical issues that users address and the questions of systematic reviews, as well as the risk of bias in the creation and interpretation of the results of systematic reviews (including interventions, diagnosis, prognosis, and etiology) [38]. We will also utilize the ROBIS Tool to assess the systematic review's risk of bias, based on the kind of research that will be included in it. It is expected that a more thorough evaluation of the systematic review's evidence level will be possible due to the complementarity of the GRADE standard and the ROBIS tool.

## Discussion

It has become widely accepted that T2DM and cognitive dysfunction are progressive diseases closely related to age, which have a high prevalence in older people and those of middle age [39, 40]. T2DM with cognitive dysfunction has undoubtedly gotten a lot of attention in the medical and health fields as a result of the emergence of a world that is becoming older and the expansion of persons with T2DM's life expectancy globally. However, the relationship between hypoglycemic events, one of the typical complications of glycemic control in individuals

suffering from T2DM, and cognitive impairment remains unclear [41]. This investigation seeks to explore the connection between hypoglycemic events and cognitive impairment, and further explore the effects of the severity, duration, and frequency of hypoglycemic events on cognitive impairment by subgroup analysis or meta-regression. We will refine and distinguish the types of cognitive dysfunction once we have sufficient data, in order to clarify the precise association between hypoglycemia and different types of cognitive dysfunction. The results of the study will facilitate the development of personalized glycemic control strategies for patients with different conditions of T2DM, and hopefully improve the prevention strategies for different types of cognitive dysfunction related to diabetes.

Many medical staff working on the clinical front line have observed that people with frequent hypoglycemic events seem to be more likely to develop cognitive impairment [42]. The association between hypoglycemia incidents and cognitive impairment in terms of dose-response hasn't been explored, though. This is very disadvantageous for the elderly population, who frequently experience hypoglycemic events. On the one hand, they do not understand the long-term and cumulative damage of hypoglycemic events to cognitive function, and on the other hand, they lack awareness of the management of hypoglycemic events, which is also an important reason for the common occurrence of hypoglycemic events in the elderly [43]. Understanding the dose-response relationship between hypoglycemic events and cognitive dysfunction not only directly affects the elaboration of precise hypoglycemic regimens, but also provides decision-making guidance for relevant medical workers and government policy agencies. Most importantly, it will help to enhance the awareness of hypoglycemia prevention in patients with T2DM, thereby reducing the cumulative damage of cognitive function caused by hypoglycemia and preventing cognitive impairment.

Hypoglycemic episodes and cognitive impairment, of course, have a complex dose-response connection that might be altered by a wide range of lifestyle factors. According to studies, a healthy lifestyle that includes things like exercise, social interaction, napping, and eating a balanced diet is linked to a slower rate of memory loss [44, 45]. One of the major factors contributing to the development of cognitive impairment in diabetics is the pervasive inflammatory condition [46]. According to studies, leading a healthy lifestyle—which includes exercise in particular—protects the brain from inflammatory states like CRP and enhances cognitive function [47]. Exercise, however, may raise the risk of hypoglycemia in diabetes individuals using insulin or insulin secretagogues, according to previous research [48, 49]. Exercise-related hypoglycemia impairs cognitive performance by inhibiting the autonomic nervous system, neuroendocrine system, and metabolic defenses (also known as counter-regulatory responses) [50]. Consequently, exercise may have a bidirectional effect on the dose-response connection between hypoglycemia events and cognitive impairment. Future research should be planned to account for or adjust for these confounders because it is yet unknown how other lifestyle factors, such as food and sleep, may affect the study's estimated effect size. Eventually, these initiatives should produce more thorough research findings and practical lifestyle guidelines or suggestions for reducing cognitive impairment in diabetics.

In conclusion, this study has the potential to address the evidence gap on the connection between hypoglycemic events and cognitive dysfunction in patients suffering from T2DM. It also has the potential to promote glycemic and cognitive management, benefiting the large population of those suffering from T2DM.

## Strengths of the proposed meta-analysis

There is currently no specific treatment plan for diabetes-related cognitive impairment, despite its high frequency. There is debate over whether hypoglycemia episodes are a major factor in

type 2 diabetes patients' cognitive impairment. It is anticipated that carrying out this study protocol will give rise to an evidence-based foundation supporting the link between hypoglycemia episodes and cognitive decline. It will also investigate the dose-response association between hypoglycemia and cognitive impairment for the first time. It is anticipated that this protocol will offer helpful recommendations for blood glucose regulation and the prevention of cognitive impairment in patients with type 2 diabetes.

## Limitations of the proposed meta-analysis

First of all, the utilization of different literature types in this study protocol, including case-control and cohort studies, may result in some heterogeneity. Second, bias may be produced and the degree of evidence may be lowered because it only comprised observational research. In addition, the study will just incorporate English and Chinese literature, which may limit the total amount of data that may be extracted. The systematic review report will discuss these as study limitations.

## Supporting information

**S1 Appendix. The PRISMA-P checklist.**
(DOCX)

**S2 Appendix. The search strategy for PubMed.**
(DOCX)

**S1 Fig. The PRISMA flow diagram of the study selection process.**
(TIF)

**S1 File.**
(XLSX)

## Acknowledgments

The study's idea and the manuscript's revision involved input from all authors. We are appreciative to teacher Ling Cheng from the College of Humanities at the Anhui University of Chinese Medicine for editing this article's language.

## Author Contributions

**Conceptualization:** Min Ye, Ai Hong Yuan.

**Formal analysis:** Min Ye, Ai Hong Yuan, Qun Wei Li.

**Investigation:** Qi Qi Yang, Qun Wei Li.

**Methodology:** Qi Qi Yang, Qun Wei Li, Yan Wei.

**Project administration:** Ai Hong Yuan.

**Resources:** Qi Qi Yang.

**Software:** Fei Yue Li.

**Supervision:** Fei Yue Li.

**Writing – original draft:** Min Ye.

**Writing – review & editing:** Ai Hong Yuan.

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
