## [Decision Letter · Decision Letter 0]

27 Oct 2023

PONE-D-23-23141Association of hypoglycemic events with cognitive impairment in patients with type 2 diabetes mellitus: Protocol for a dose-response meta-analysisPLOS ONE

Dear Dr. Ye,

Thank you for submitting your manuscript to PLOS ONE. After careful consideration, we feel that it has merit but does not fully meet PLOS ONE’s publication criteria as it currently stands. Therefore, we invite you to submit a revised version of the manuscript that addresses the points raised during the review process.

We look forward to receiving your revised manuscript.

Kind regards,

Muhammad Shahzad Aslam, Ph.D.,M.Phil., Pharm-D

Academic Editor

PLOS ONE

Reviewers' comments:

Reviewer's Responses to Questions

**Comments to the Author**

1. Does the manuscript provide a valid rationale for the proposed study, with clearly identified and justified research questions?

Reviewer #1: Yes

Reviewer #2: Yes

Reviewer #3: Yes

Reviewer #4: Yes

2. Is the protocol technically sound and planned in a manner that will lead to a meaningful outcome and allow testing the stated hypotheses?

Reviewer #1: Yes

Reviewer #2: Yes

Reviewer #3: Yes

Reviewer #4: Yes

3. Is the methodology feasible and described in sufficient detail to allow the work to be replicable?

Reviewer #1: Yes

Reviewer #2: No

Reviewer #3: Yes

Reviewer #4: Yes

4. Have the authors described where all data underlying the findings will be made available when the study is complete?

Reviewer #1: Yes

Reviewer #2: Yes

Reviewer #3: Yes

Reviewer #4: Yes

5. Is the manuscript presented in an intelligible fashion and written in standard English?

Reviewer #1: Yes

Reviewer #2: Yes

Reviewer #3: Yes

Reviewer #4: Yes

6. Review Comments to the Author

You may also provide optional suggestions and comments to authors that they might find helpful in planning their study.

Reviewer #1: Dear authors,

I have reviewed your work and appreciate your meticulous work on this subject. Overall, your protocol seems well designed and comprehensive. The literature review and data extraction methods appear to be comprehensive and robust, appropriate to the purpose of the study.

I suggest you take into consideration my suggestions below:

1-You can further expand the keywords used for the literature search. (For example: neurodegeneration,neurodegenerative disease, neurocognitive disorder, neuropsychiatric disorder) This will help researchers screen for all relevant studies.

2-You can make the information used for data extraction more detailed. Definition of cognitive impairment, Cognitive impairment measurements, etc.

3-You can define the variables to be used for dose-response analysis in more detail.

I wish you a successful completion of your work.

Reviewer #2: This protocol details the methodology to be followed for a systematic review and meta-analysis on hypoglycemic events and cognitive impairment in patients with type 2 diabetes mellitus. Ye et al. present an interesting protocol, well-structured and with a methodologic plan of interest. This type of study is needed in this population, so the protocol is relevant.

However, there are several comments that should be noted and addressed:

a. Eligibility criteria. Detail inclusion criteria for exposure, such as definition for hypoglycemic events or assessment methods. Furthermore, specify criteria for outcome (cognitive impairment), i.e., assessment methods, definition, etc.

b. Line 96. Specify the type of observational studies to be included.

c. Line 97. “Individuals with a diagnosis of T2DM will be incorporated”. Will only studies that report medical diagnosis be included or could they be self-reported? Please specify. Also, will the participants included be adults? Please specify and indicate age range.

d. Line 103. Please delete criterion 1. This is a repeated criterion that has already been established as an inclusion criterion.

e. Line 131.

f. Line 153. Will a minimum number of studies be considered for meta-analysis? Will different meta-analyses be performed depending on the study design? Please detail.

g. Line 157. Provide the interpretation of I2 according to the percentages of heterogeneity and its respective reference.

h. Line 165. Will a minimum of studies be considered for meta-regression analyses? Consider Cochrane recommendations.

i. For subgroup and meta-regression analyses consider BMI status and physical activity levels. Also, add these variables for data extraction.

j. Lines 171-172. Please provide more detail on the possible dose-response curve analysis to be performed.

k. Lines 174-175. Will a minimum of studies be considered for publication bias? Consider Sterne et al. (2011) recommendations. Moreover, for the linear regression approach, what will be the p-value considered significant?

l. Lines 177-178. Specify the characteristics of the levels of evidence. Detail the levels of evidence possible to be graded according to GRADE. What does high, medium, low and very low mean?

m. I think it would be useful for the authors to discuss the potential role of exercise, sleep, and other lifestyle behaviors in the association between hypoglycemic events and cognitive decline in patients with type 2 diabetes mellitus. In this sense, manuscripts with an integrative and translational character should be included in the discussion, such as doi: https://doi.org/10.1186/s12902-019-0402-3; doi: https://doi.org/10.1586/eem.10.78; doi: https://doi.org/10.4093/dmj.2022.0007; doi: https://doi.org/10.1136/bmj-2022-072691; and doi: http://dx.doi.org/10.1136/bjsports-2022-106355

Reviewer #3: Comments to the authors

Abstract

• “The connection that exists between hypoglycemic occurrences and the likelihood of cognitive impairment will be investigated through an organized review of multiple databases.” It is better to include a sentence summarising the inclusion criteria: "Studies that ........ were included".

• “Both Chinese and English literature…” Remove this from the summary.

Introduction

• “…normal individuals” Change from normal to healthy

• “By constructing a diabetic rat model, Seok Joon Won…”. Better: "In animal models, it has been observed that..."

• “This investigation seeks to update the systematic review and meta-analysis by including the latest relevant studies”. Better “This systematic review and meta-analyses aimed to…”

Methods and materials

• If it is a meta-analysis of observational studies, the MOOSE guideline should also be included.

• “All observational studies published in Chinese or English will be included”. Why not include other languages? Even if authors only know English and Chinese, there are translation tools that allow studies in other languages to be included.

• “The study has no restrictions on sex, race, duration of diabetes, or severity of diabetes among the participants” But is the severity of diabetes going to be controlled in some way?

• “…likelihood of cognitive dysfunction will be included”. Is there a way to determine if there is cognitive disorder? It is important to establish this a priori before the study.

• Inclusion criteria – “Studies not published in Chinese or English” and “Participants with gestational diabetes”. Exclusion criteria are not the opposite of inclusion criteria. Exclusion criteria are those that exclude participants or studies from inclusion even though they meet the inclusion criteria.

• Search strategy – I suggest this section be placed before the eligibility criteria.

• Data extraction – “the primary author” ¿first author? “period of publication” ¿year of publication? “type of diabetes mellitus” Isn't it just type 2 diabetes mellitus? “type of cognitive impairment” The type of cognitive dysfunction that can be included needs to be defined beforehand.

• “Evaluation of literature quality (publication bias)” – First, how many authors did the publication bias assessment? Secondly, I do not recommend putting publication bias. Publication bias generally refers to the Egger test and the funnel plot for meta-analysis.

• Data synthesis – First, the authors should rate the heterogeneity, for example, as not important, moderate, substantial, and considerable. Second, the p-value of heterogeneity should be assessed. Third, why did you choose the value of 30% or more to do a random-effects meta-analysis? Why not 50%? Or why not if p is < 0.05? I suggest doing the random effects meta-analysis, studies with some heterogeneity in the population and its characteristics will be included, and fixed effects meta-analyses usually give narrower confidence intervals, which can be misleading in some contexts.

• Publication bias – And what p-value is used as a threshold to consider that there is publication bias?

• Grading the quality of evidence – It is proposed to explain it a little more (not much, but a little more, yes)

Reviewer #4: Congratulations to the authors. This is a well-written and constructed manuscript. I enjoyed learning about the study. I believe it is worthy of publication.

Some minor changes:

1. Abstract

- I suggest including the specific study design (systematic review and meta-analysis protocol).

- I recommend including both that this protocol will be adhered to the Preferred Reporting Items for Systematic Reviews and Meta-Analyses Protocols and the meta-analysis will be guided by the Cochrane Collaboration Handbook recommendations.

- Line 28: I suggest specifying the way in which the inconsistences will be solved.

2. Introduction

- Great

- Objective: The objective of this manuscript is not to explore the cumulative effect of the frequency of hypoglycemia events on cognitive function, but to establish the methodological structure (protocol) in order to explore, through systematic review and meta-analysis, the cumulative effect of the frequency of hypoglycemia events on cognitive function. Reconsider.

3. Methods and materials

- Line 89. The systematic review should also be adhered to the Meta-analysis Of Observational Studies in Epidemiology (MOOSE) statement.

- Line 96. Could the author please state the reason for the exclusion of experimental studies?

- Line 156: It is suggested to indicate the range of heterogeneity based on I2 (not important, moderate, substantial, and considerable)

- Will the authors consider the corresponding p-value?

4. Discussion

- Line 190: The authors state that “This investigation seeks to explore the connection between hypoglycemic events and cognitive impairment”. However, they pretend to “clarify the precise association between hyperglycemia and different types of cognitive dysfunction”. Is this correct?

7. PLOS authors have the option to publish the peer review history of their article (what does this mean?). If published, this will include your full peer review and any attached files.

Reviewer #1: No

Reviewer #2: No

Reviewer #3: **Yes: **Carlos Pascual-Morena; Irene Martínez-García

Reviewer #4: No

---

## [Author Response · Author response to Decision Letter 0]

6 Nov 2023

Response to journal's requirements

Ans: We appreciate you reminding us. We have made the necessary formatting adjustments in accordance with PLOS ONE style guidelines, and we trust that the manuscript now satisfies your criteria.

2.We note that the grant information you provided in the ‘Funding Information’ and ‘Financial Disclosure’ sections do not match. When you resubmit, please ensure that you provide the correct grant numbers for the awards you received for your study in the ‘Funding Information’ section.

Ans: We appreciate you reminding us. We have reviewed the funding information portion of the manuscript and determined that the data it contains is correct. We apologize for any trouble this has given you with your work and will be updating the incorrect information in the online submission form.

3.Your ethics statement should only appear in the Methods section of your manuscript. If your ethics statement is written in any section besides the Methods, please move it to the Methods section and delete it from any other section. Please ensure that your ethics statement is included in your manuscript, as the ethics statement entered into the online submission form will not be published alongside your manuscript.

Ans: We appreciate your proposal, and as per it, we have relocated the ethical statement to the manuscript methodology section and deleted it from its previous location.

 

Response to Reviewer 1's Comments

Firstly, we would like to express our sincere thanks for your constructive comments and positive response. The revised version of the manuscript has been significantly modified based on your comments. The revised parts are marked in red font. We hope you are satisfied with our revisions.

1. You can further expand the keywords used for the literature search. (For example: neurodegeneration, neurodegenerative disease, neurocognitive disorder, neuropsychiatric disorder) This will help researchers screen for all relevant studies.

Ans: We have broadened relevant search terms based on your suggestions to completely choose relevant papers, as shown in lines 100-115 and S2 Appendix.

2. You can make the information used for data extraction more detailed. Definition of cognitive impairment, Cognitive impairment measurements, etc.

Ans: Thanks to your advice, we have modified the required data extraction information, making our scheme more complete and practicable, as shown in lines 148-160.

3. You can define the variables to be used for dose-response analysis in more detail.

Ans: We specified possible variables for dose-response analysis based on your suggestions, such as frequency of hypoglycemia, severity of hypoglycemia, duration of hypoglycemia, and incidence of cognitive impairment, to provide a clearer picture of our study protocol. For more information, see lines 154-160 and 197-202.

 

Response to Reviewer 2's Comments

Firstly, we would like to express our sincere thanks for your constructive comments and positive response. The revised version of the manuscript has been significantly modified based on your comments. The revised parts are marked in red font. We hope you are satisfied with our revisions.

1.Eligibility criteria. Detail inclusion criteria for exposure, such as definition for hypoglycemic events or assessment methods. Furthermore, specify criteria for outcome (cognitive impairment), i.e., assessment methods, definition, etc. 

Ans: Thanks for your valuable advice, we have added the assessment methods for hypoglycemic events and cognitive impairment in the inclusion criteria section according to your suggestion. See lines 123-131 for details.

2. Line 96. Specify the type of observational studies to be included.

Ans: We have clarified the study types of the included observational research, such as cross-sectional studies, case-control studies, cohort studies, and so on, in response to your recommendation. For more information, see lines 118-119 of the text.

3.Line 97. “Individuals with a diagnosis of T2DM will be incorporated”. Will only studies that report medical diagnosis be included or could they be self-reported? Please specify. Also, will the participants included be adults? Please specify and indicate age range.

Ans: Thank you for your extremely useful advice. We clarified the characteristics of subjects in related studies based on your ideas. To assure the scientific validity and rigor of the trial results, we demand that all subjects be medically diagnosed with type 2 diabetes and be adults, that is, above the age of 18. Details can be found in lines 120-121.

4.Line 103. Please delete criterion 1. This is a repeated criterion that has already been established as an inclusion criterion.

Ans: Thank you for your wise counsel. This is the result of our carelessness. We removed criterion 1 based on your advice.

5. Line 153. Will a minimum number of studies be considered for meta-analysis? Will different meta-analyses be performed depending on the study design? Please detail.

Ans: Thank you for your valuable advice and in accordance with the Cochrane Handbook for Systematic Reviews, we have stipulated that meta-analyses will be performed only if there are sufficient studies (≥2) with similar definitions of exposure and outcome of interest, and only descriptive analyses will be performed if the number of studies is insufficient. When the number of studies and relevant data are sufficiently rich, we will conduct subgroup analysis according to different study types, such as cross-sectional studies, case-control studies, cohort studies, etc. See lines 176-177 and 190 of the text for details.

6. Line 157. Provide the interpretation of I2 according to the percentages of heterogeneity and its respective reference.

Ans：We have updated the explanation of I2 for the proportion of heterogeneity and its respective reference, as you requested. Details can be found on lines 181-186.

7. Line 165. Will a minimum of studies be considered for meta-regression analyses? Consider Cochrane recommendations.

Ans: Thank you for your insightful advice. According to the Cochrane Handbook for Systematic Reviews of Interventions, the minimum number of studies for meta-regression analysis is ten. Look at lines 192-193.

8. For subgroup and meta-regression analyses consider BMI status and physical activity levels. Also, add these variables for data extraction.

Ans: Thank you for your helpful suggestions. We completely agree with your points of view and will include BMI status and physical activity levels in the data extraction, subgroup analysis, and meta-regression, making our study more valuable. For more information, see lines 152 and 189.

9. Lines 171-172. Please provide more detail on the possible dose-response curve analysis to be performed.

Ans: We have added possible exposure factors and outcome measures that may be utilized for dose-response analyses to the dose-response curve analysis section based on your recommendation. See lines 197-202 for more information.

10. Lines 174-175. Will a minimum of studies be considered for publication bias? Consider Sterne et al. (2011) recommendations. Moreover, for the linear regression approach, what will be the p-value considered significant?

Ans: Thank you for your advice. We concluded that the minimal number of literature with publication bias is ten based on your proposal and relevant literature. At the same time, a P value of less than 0.05 for Egger's regression asymmetry test is considered severe publication bias. For more information, see lines 203-206.

11. Lines 177-178. Specify the characteristics of the levels of evidence. Detail the levels of evidence possible to be graded according to GRADE. What does high, medium, low and very low mean?

Ans: We appreciate your recommendations, which helped us clarify the characteristics of the evidence level and further explain the GRADE evidence level. For more information, see lines 207-212.

12. I think it would be useful for the authors to discuss the potential role of exercise, sleep, and other lifestyle behaviors in the association between hypoglycemic events and cognitive decline in patients with type 2 diabetes mellitus. In this sense, manuscripts with an integrative and translational character should be included in the discussion, such as doi: https://doi.org/10.1186/s12902-019-0402-3; doi: https://doi.org/10.1586/eem.10.78; doi: https://doi.org/10.4093/dmj.2022.0007; doi: https://doi.org/10.1136/bmj-2022-072691; and doi: http://dx.doi.org/10.1136/bjsports-2022-106355

Ans: Thank you for your detailed constructing proposal. We read the appropriate literature thoroughly and gained greatly. Lifestyle influences the link between hypoglycemic occurrences and cognitive impairment, which has crucial implications for future studies in this area. We have added relevant content to the discussion section based on your recommendation. Details can be found in lines 245-261.

 

Response to Reviewer 3's Comments

Firstly, we would like to express our sincere thanks for your constructive comments and positive response. The revised version of the manuscript has been significantly modified based on your comments. The revised parts are marked in red font. We hope you are satisfied with our revisions.

1.“The connection that exists between hypoglycemic occurrences and the likelihood of cognitive impairment will be investigated through an organized review of multiple databases.” It is better to include a sentence summarising the inclusion criteria: "Studies that ........ were included".

Ans: Thank you for your proposal; we have provided the summary discourse of the literature inclusion criteria based on it. See lines 25-26.

2. “Both Chinese and English literature…” Remove this from the summary.

Ans: Thank you for your important advice; we have removed "Both Chinese and English literature..." following your recommendation.

3. “…normal individuals” Change from normal to healthy

Ans: We modified "normal" to "healthy" based on your advice. See line 56 for further information.

4. “By constructing a diabetic rat model, Seok Joon Won…”. Better: "In animal models, it has been observed that..."

Ans: Thank you for your helpful advice. We changed the sentence structure based on your ideas. For more information, see line 67.

5. “This investigation seeks to update the systematic review and meta-analysis by including the latest relevant studies”. Better “This systematic review and meta-analyses aimed to…”

Ans: Thank you for your valuable suggestions. We have revised the sentence structure according to your suggestions. See lines 85-89 for details.

6. If it is a meta-analysis of observational studies, the MOOSE guideline should also be included.

Ans: Thank you for your valuable suggestions, and as indicated in the methodology section, we will report our work in compliance with the MOOSE guidelines. Details can be found on lines 95-97.

7.“All observational studies published in Chinese or English will be included”. Why not include other languages? Even if authors only know English and Chinese, there are translation tools that allow studies in other languages to be included.

Ans: Thank you for your proposal; we studied it carefully and ultimately chose to incorporate as many literatures in diverse languages as feasible, resulting in more scientifically robust findings. As indicated in lines 102 and 119, we added the target retrieval database and made the required adjustments in the body section.

8. “The study has no restrictions on sex, race, duration of diabetes, or severity of diabetes among the participants” But is the severity of diabetes going to be controlled in some way?

Ans：Thank you for your beneficial suggestions; we also considered patient safety when preparing the manuscript. As a result, we required that relevant disorders be diagnosed by professional physicians or competent healthcare institutions, and that patients receive counsel and guidance from professional physicians throughout the trial. At the same time, because this meta-analysis is a secondary review of the original study's data, it does not raise ethical or patient safety concerns. We did not limit the severity of diabetes in order to include as many relevant research as possible. If the data allow, we will undertake subgroup analyses based on diabetes severity to investigate the effect of diabetes severity on the connection between hypoglycemia and cognitive impairment.

9. “…likelihood of cognitive dysfunction will be included”. Is there a way to determine if there is cognitive disorder? It is important to establish this a priori before the study.

Ans: This is a critical and serious issue. Before creating this meta-analysis protocol, we conducted a pre-search of individual databases, and the majority of studies investigating the association between hypoglycemia and cognitive impairment performed cognitive function tests before individuals were enrolled. Subjects with impaired cognition were screened out because they did not match the inclusion criteria for these investigations since hypoglycemia was specified as an exposure factor and cognitive impairment was set as an outcome measure. This study was a meta-analysis of the findings from these studies, and cognitive impairment should be mentioned as an outcome indicator and its diagnostic criteria in the inclusion criteria. This was an oversight in the manuscript's preparation, which is now detailed in the inclusion criteria you requested, as indicated in lines 126-131.

10. Inclusion criteria – “Studies not published in Chinese or English” and “Participants with gestational diabetes”. Exclusion criteria are not the opposite of inclusion criteria. Exclusion criteria are those that exclude participants or studies from inclusion even though they meet the inclusion criteria.

Ans：Thank you for the suggestions. We removed the above two improper exclusion criteria based on your feedback.

11. Search strategy – I suggest this section be placed before the eligibility criteria.

Ans：Thanks for your suggestion, we have adjusted the order of the relevant content according to your suggestion, see lines 100 and 116 for details.

12. Data extraction – “the primary author” ¿first author? “period of publication” ¿year of publication? “type of diabetes mellitus” Isn't it just type 2 diabetes mellitus? “type of cognitive impairment” The type of cognitive dysfunction that can be included needs to be defined beforehand.

Ans：Thank you for your advice. We regret for the manuscript's confusing and erroneous description, and we have edited, eliminated, and replaced the appropriate text based on your suggestions. For more information, see lines 150-160.

13. “Evaluation of literature quality (publication bias)” – First, how many authors did the publication bias assessment? Secondly, I do not recommend putting publication bias. Publication bias generally refers to the Egger test and the funnel plot for meta-analysis.

Ans: Thank you for your advice. As per your suggestion, we have clarified in the publication that the assessment of the quality of the literature was conducted by two investigators, and any disagreements were resolved by a third investigator. We have removed the inappropriate phrase "publication bias" according to your suggestion. See lines 161-164 for details.

14. Data synthesis – First, the authors should rate the heterogeneity, for example, as not important, moderate, substantial, and considerable. Second, the p-value of heterogeneity should be assessed. Third, why did you choose the value of 30% or more to do a random-effects meta-analysis? Why not 50%? Or why not if p is < 0.05? I suggest doing the random effects meta-analysis, studies with some heterogeneity in the population and its characteristics will be included, and fixed effects meta-analyses usually give narrower confidence intervals, which can be misleading in some contexts.

Ans：Thank you for your valuable suggestions, which helped us clarify the heterogeneity rating. Initially, I2≥30% was chosen for random effect model analysis in order to estimate the effect size more conservatively, however after reviewing your idea, we believe that the previous consideration is unnecessary and erroneous. From this point on, we use the random effects model when either P < 0.1 or I2 ≥ 50%. Details can be found on lines 181-186.

15. Publication bias – And what p-value is used as a threshold to consider that there is publication bias?

Ans: Thanks for your suggestion, we have clarified the threshold of P value in the main text, when a P value < 0.05 is considered to have significant publication bias. See lines 175-186 for details.

16. Grading the quality of evidence – It is proposed to explain it a little more (not much, but a little more, yes)

Ans：Thanks for your guidance, we have elaborated the grading of the quality of evidence. See lines 207-212 for details. 

Response to Reviewer 4's Comments

Firstly, we would like to express our sincere thanks for your constructive comments and positive response. The revised version of the manuscript has been significantly modified based on your comments. The revised parts are marked in red font. We hope you are satisfied with our revisions.

1.I suggest including the specific study design (systematic review and meta-analysis protocol).

Ans：We have added specific study designs to the abstract section as suggested by you. See line 23 for details.

2. I recommend including both that this protocol will be adhered to the Preferred Reporting Items for Systematic Reviews and Meta-Analyses Protocols and the meta-analysis will be guided by the Cochrane Collaboration Handbook recommendations.

Ans: We appreciate your insightful counsel, which we will take into consideration as we adhere to the Cochrane Collaboration Handbook recommendations and the preferred reporting items for the Systematic Reviews and Meta-analyses Protocols. We provide further details on this in the section on Methods and Materials, and in the Supplementary material, we include the updated PRISMA-P checklist. For details, see lines 92–99.

3. Line 28: I suggest specifying the way in which the inconsistences will be solved.

Ans: We appreciate your insightful counsel, and as per it, we have clarified that any discrepancies in the literature screening procedure were addressed by a different researcher. See line 33.

4.Objective: The objective of this manuscript is not to explore the cumulative effect of the frequency of hypoglycemia events on cognitive function, but to establish the methodological structure (protocol) in order to explore, through systematic review and meta-analysis, the cumulative effect of the frequency of hypoglycemia events on cognitive function. Reconsider.

Ans: Thank you for your constructive suggestions. We have revised the inappropriate expression according to your suggestions. See lines 86-89 for details.

5.Line 89. The systematic review should also be adhered to the Meta-analysis Of Observational Studies in Epidemiology (MOOSE) statement.

Ans: Thank you for your valuable suggestions. We have modified the relevant content according to your suggestions, which makes our scheme look more rigorous. See lines 96-97 for details.

6. Line 96. Could the author please state the reason for the exclusion of experimental studies?

Ans: Thank you for your question. First of all, this study aims to explore the relationship between hypoglycemic events (exposure factors) and cognitive impairment (outcome measures) in the natural state, and does not involve exploring the role of artificial interventions such as treatment measures. Second, hypoglycemic events are highly harmful to the human body and have unpredictable characteristics, so it is almost impossible to recruit subjects with stable hypoglycemia for experimental research due to ethical considerations. The current experimental studies on the relationship between hypoglycemic events and cognitive impairment are almost based on the construction of hypoglycemic animal models. Finally, this meta-analysis needs to use data such as odds ratio (OR), risk ratio (RR), hazard ratio (HR), and 95% confidence intervals (CI), which are not available in experimental studies. Therefore, only observational studies were included in this study.

7.Line 156: It is suggested to indicate the range of heterogeneity based on I2 (not important, moderate, substantial, and considerable). Will the authors consider the corresponding p-value?

Ans： Thank you for your proposal. We used the range of I2 values to grade the heterogeneity in accordance with your suggestion and the Cochrane Handbook. We also took into consideration that a P value < 0.1 was deemed to be significant heterogeneity. We have meticulously edited and illustrated the primary content. For more information, see lines 181–186.

8. Line 190: The authors state that “This investigation seeks to explore the connection between hypoglycemic events and cognitive impairment”. However, they pretend to “clarify the precise association between hyperglycemia and different types of cognitive dysfunction”. Is this correct?

Ans：Thank you for your careful review and finding this spelling error. We are very sorry for our carelessness. According to your suggestion, we have modified the text. See line 227 for details.

---

## [Decision Letter · Decision Letter 1]

23 Nov 2023

PONE-D-23-23141R1Association of hypoglycemic events with cognitive impairment in patients with type 2 diabetes mellitus: Protocol for a dose-response meta-analysisPLOS ONE

Dear Dr. Yuan,

Thank you for submitting your manuscript to PLOS ONE. After careful consideration, we feel that it has merit but does not fully meet PLOS ONE’s publication criteria as it currently stands. Therefore, we invite you to submit a revised version of the manuscript that addresses the points raised during the review process.

 Please submit your revised manuscript by Jan 07 2024 11:59PM. If you will need more time than this to complete your revisions, please reply to this message or contact the journal office at plosone@plos.org. Please include the following items when submitting your revised manuscript:A rebuttal letter that responds to each point raised by the academic editor and reviewer(s). You should upload this letter as a separate file labeled 'Response to Reviewers'.A marked-up copy of your manuscript that highlights changes made to the original version. You should upload this as a separate file labeled 'Revised Manuscript with Track Changes'.An unmarked version of your revised paper without tracked changes. You should upload this as a separate file labeled 'Manuscript'.If applicable, we recommend that you deposit your laboratory protocols in protocols.io to enhance the reproducibility of your results. Protocols.io assigns your protocol its own identifier (DOI) so that it can be cited independently in the future. For instructions see: https://journals.plos.org/plosone/s/submission-guidelines#loc-laboratory-protocols. Additionally, PLOS ONE offers an option for publishing peer-reviewed Lab Protocol articles, which describe protocols hosted on protocols.io. Read more information on sharing protocols at https://plos.org/protocols?utm_medium=editorial-email&utm_source=authorletters&utm_campaign=protocols.

We look forward to receiving your revised manuscript.

Kind regards,

Muhammad Shahzad Aslam, Ph.D.,M.Phil., Pharm-D

Academic Editor

PLOS ONE

Journal Requirements:

Reviewers' comments:

Reviewer's Responses to Questions

**Comments to the Author**

1. Does the manuscript provide a valid rationale for the proposed study, with clearly identified and justified research questions?

Reviewer #1: Yes

Reviewer #2: Yes

Reviewer #3: Yes

Reviewer #4: Yes

2. Is the protocol technically sound and planned in a manner that will lead to a meaningful outcome and allow testing the stated hypotheses?

Reviewer #1: Yes

Reviewer #2: Yes

Reviewer #3: Yes

Reviewer #4: Yes

3. Is the methodology feasible and described in sufficient detail to allow the work to be replicable?

Reviewer #1: Yes

Reviewer #2: Yes

Reviewer #3: Yes

Reviewer #4: Yes

4. Have the authors described where all data underlying the findings will be made available when the study is complete?

Reviewer #1: Yes

Reviewer #2: Yes

Reviewer #3: Yes

Reviewer #4: Yes

5. Is the manuscript presented in an intelligible fashion and written in standard English?

Reviewer #1: Yes

Reviewer #2: Yes

Reviewer #3: Yes

Reviewer #4: Yes

6. Review Comments to the Author

You may also provide optional suggestions and comments to authors that they might find helpful in planning their study.

Reviewer #1: Dear authors

I have carefully considered your responses to my requests for revisions and am pleased to note that your changes have both broadened and deepened the study. The expansion of the keywords has made the literature search more comprehensive, while the elaboration of the data extraction information has significantly increased the transparency and credibility of the study. In addition, the more detailed definition of the variables to be used for the dose-response analysis strengthens the methodological integrity of the study and contributes to the interpretation of the results.

Thank you for your hard work and diligence in this process.

Reviewer #2: The authors adrressed satisfatorily all the commentts and the manuscript can now be accepted for publication

Reviewer #3: Comments to the authors

Most of the comments have been dealt with satisfactorily, but I still have a few issues that I feel should be addressed.

• “There will be four categories for heterogeneity: not important (0%-40%), moderate (30%-60%), substantial (50%-90%) and considerable (75%-90%). The fixed effect model will be applied for combined analysis if the I2 value is less than 50%. A random effect model will be applied when P < 0.1 or I2 > 50%[32, 33]. Subgroup analysis and other techniques will be used to investigate sources of clinical or methodological heterogeneity when I2 > 75%.” - I suggest that considerable is 75-100%. I also suggest including subgroup studies even if there is no detected statistically significant heterogeneity.

• Publication bias – Why is p < 0.10 not used to assume publication bias? It is the most commonly used value.

Reviewer #4: Congratulations to the authors. This is a well-written and constructed manuscript. I enjoyed learning about the study. I believe it is worthy of publication.

Some minor changes:

- Line 85: I propose to rephrase the objective. I suggest the following: "This manuscript aims to establish a methodological structure (protocol) to explore the cumulative effect of frequency of hypoglycemic events on cognitive function and further explore the dose-response relationship through systematic review and meta-analysis, which aims to update the association between hypoglycemia and cognitive impairment by including the latest relevant studies."

- Could the authors please cite the MOOSE statement?

7. PLOS authors have the option to publish the peer review history of their article (what does this mean?). If published, this will include your full peer review and any attached files.

Reviewer #1: No

Reviewer #2: No

Reviewer #3: **Yes: **Carlos Pascual-Morena; Irene Martínez-García

Reviewer #4: No

---

## [Author Response · Author response to Decision Letter 1]

28 Nov 2023

Response to journal's requirements

Ans: I appreciate your kind reminder. We have conducted a thorough investigation of each reference once more by your recommendation. To standardize the reference format, all references were downloaded from PubMed and entered into the Endnote program. There is no issue number for references 7, 12, 13, 14, 39, and 43, and there is no digital object identifier number for reference 47 since it is not included in the original journal. Simultaneously, we examined all the references and were happy to see that none of the withdrawn publications had been cited. Once again, I appreciate your thorough review. 

Response to Reviewer 3's Comments

Firstly, we would like to express our sincere thanks for your constructive comments and positive response. The revised version of the manuscript has been significantly modified based on your comments. The revised parts are marked in red font. We hope you are satisfied with our revisions.

1. “There will be four categories for heterogeneity: not important (0%-40%), moderate (30%-60%), substantial (50%-90%) and considerable (75%-90%). The fixed effect model will be applied for combined analysis if the I2 value is less than 50%. A random effect model will be applied when P < 0.1 or I2 > 50%[32, 33]. Subgroup analysis and other techniques will be used to investigate sources of clinical or methodological heterogeneity when I2 > 75%.” - I suggest that considerable is 75-100%. I also suggest including subgroup studies even if there is no detected statistically significant heterogeneity. 

Ans: Regards for your insightful recommendations. About the I2 value classification, while there are currently a variety of versions, following a collaborative discussion among all authors, we consider your recommendations to be more feasible and scientifically sound, so we modify the considerable heterogeneity to 75–100% by your recommendations. We made adjustments for assertions that were not accurate. That is, regardless of whether statistically significant heterogeneity is found, subgroup analysis and other workable analytical techniques will be used to investigate clinical heterogeneity if there is enough literature. Details are shown in lines 182-187.

2. Publication bias – Why is p < 0.10 not used to assume publication bias? It is the most commonly used value. 

Ans: We appreciate you informing us that two types of publication bias P-values are currently in use: 0.05 and 0.1. Since 0.1 is more rigorous and widely used, we have adjusted the P-value to 0.1 per your recommendation. For more information, see lines 206-207. 

Response to Reviewer 4's Comments

Firstly, we would like to express our sincere thanks for your constructive comments and positive response. The revised version of the manuscript has been significantly modified based on your comments. The revised parts are marked in red font. We hope you are satisfied with our revisions.

1. Line 85: I propose to rephrase the objective. I suggest the following: "This manuscript aims to establish a methodological structure (protocol) to explore the cumulative effect of frequency of hypoglycemic events on cognitive function and further explore the dose-response relationship through systematic review and meta-analysis, which aims to update the association between hypoglycemia and cognitive impairment by including the latest relevant studies.".

Ans: We appreciate your patient advice, and we have adapted the research purpose in light of your recommendations, which improves the overall quality and scientific appearance of our program. For information, see lines 85–89.

2. Could the authors please cite the MOOSE statement?

Ans: We appreciate you reminding us and apologize for this error. As per your recommendation, we have now included the pertinent references of the MOOSE statement. For information, see lines 96–97 and 356–359.

---

## [Editor Report · Decision Letter 2]

5 Dec 2023

PONE-D-23-23141R2Association of hypoglycemic events with cognitive impairment in patients with type 2 diabetes mellitus: Protocol for a dose-response meta-analysisPLOS ONE

Dear Dr. Yuan,

Thank you for submitting your manuscript to PLOS ONE. After careful consideration, we feel that it has merit but does not fully meet PLOS ONE’s publication criteria as it currently stands. Therefore, we invite you to submit a revised version of the manuscript that addresses the points raised during the review process.

1) The researcher can disagree with the comment given by reviewer.

For example, Reviewer stated that "2. Publication bias – Why is p < 0.10 not used to assume publication bias? It is the

most commonly used value.

Ans: We appreciate you informing us that two types of publication bias P-values are

currently in use: 0.05 and 0.1. Since 0.1 is more rigorous and widely used, we have

adjusted the P-value to 0.1 per your recommendation. For more information, see lines

206-207.

As an editor, suggest it to keep statistical significance in scientific research is p < 0.05. I have given my details below. Moreover, Please explain in detail the Publication bias. The current has no justification and explanation.

The commonly used threshold for statistical significance in scientific research is p < 0.05, which means that the observed results are unlikely to have occurred by chance alone. However, the use of p < 0.10 to assume publication bias is not a standard practice. The threshold of p < 0.05 is widely accepted in the scientific community and is considered a standard level of significance.

The rationale behind using a stricter threshold for publication bias assessment is to reduce the risk of false positives or Type I errors. Setting a more lenient threshold, such as p < 0.10, increases the likelihood of incorrectly concluding that there is publication bias when it may not actually be present.

When assessing publication bias, researchers often use various statistical tests and graphical methods, such as funnel plots, Egger's test, and Begg's test. These tools help evaluate whether there is a systematic relationship between study precision and effect size, which could indicate the presence of publication bias. 

2) Please Revisit the exclusion criteria. I am giving some suggestion. It is not mandatory to include all suggestion. yet your study exclusion criteria is important

Study Design:

Exclude studies that are not primary research articles (e.g., reviews, editorials, commentaries).

Exclude observational studies lacking a clear dose-response relationship or studies not reporting relevant data for a dose-response analysis.

Exclude studies with insufficient methodological detail or poor quality (e.g., high risk of bias, inadequate statistical analysis).

Population:

Exclude studies that focus exclusively on populations other than patients with type 2 diabetes mellitus.

Exclude studies with mixed populations where data on patients with type 2 diabetes cannot be extracted separately.

Exclude studies with comorbid conditions that could independently affect cognitive function (e.g., neurodegenerative diseases other than diabetes-related cognitive impairment).

Intervention/Exposure:

Exclude studies that do not report hypoglycemic events or provide insufficient information on the exposure.

Exclude studies where the dose or severity of hypoglycemic events is not clearly defined.

Exclude studies where the exposure assessment does not align with the objectives of the dose-response meta-analysis.

Outcome:

Exclude studies that do not report cognitive impairment as an outcome.

Exclude studies that use different definitions or measurements of cognitive impairment.

Exclude studies with insufficient data for the dose-response relationship or studies not reporting relevant effect measures.

Publication Characteristics:

Exclude studies published in languages other than those your team can review.

Exclude studies with incomplete or inaccessible data, and those without sufficient details to conduct a dose-response analysis.

Duration and Follow-up:

Exclude studies with a duration that is too short to capture meaningful cognitive changes.

Exclude studies with inadequate follow-up periods or those not reporting relevant follow-up data.

Publication Date:

Consider excluding older studies if there have been significant advancements in the understanding of diabetes-related cognitive impairment over time.

Intervention Heterogeneity:

Exclude studies with significant heterogeneity in interventions or treatment protocols that may impact the ability to conduct a meaningful dose-response analysis.

3) Please give two heading study strength and study limitation and explain in depth. 

4) The rational to use Grading the quality of evidence is not clear. Explain in depth. It is suggestion compare GRADE with AMSTAR 2 (A MeaSurement Tool to Assess systematic Reviews), AHRQ (Agency for Healthcare Research and Quality) Methods Guide and ROBIS (Risk of Bias in Systematic Reviews) Tool.

We look forward to receiving your revised manuscript.

Kind regards,

Muhammad Shahzad Aslam, Ph.D.,M.Phil., Pharm-D

Academic Editor

PLOS ONE

Journal Requirements:

Additional Editor Comments:

1) The researcher can disagree with the comment given by reviewer.

For example, Reviewer stated that "2. Publication bias – Why is p < 0.10 not used to assume publication bias? It is the

most commonly used value.

Ans: We appreciate you informing us that two types of publication bias P-values are

currently in use: 0.05 and 0.1. Since 0.1 is more rigorous and widely used, we have

adjusted the P-value to 0.1 per your recommendation. For more information, see lines

206-207. 

As an editor, suggest it to keep statistical significance in scientific research is p < 0.05. I have given my details below. Moreover, Please explain in detail the Publication bias. The current has no justification and explanation.

The commonly used threshold for statistical significance in scientific research is p < 0.05, which means that the observed results are unlikely to have occurred by chance alone. However, the use of p < 0.10 to assume publication bias is not a standard practice. The threshold of p < 0.05 is widely accepted in the scientific community and is considered a standard level of significance.

The rationale behind using a stricter threshold for publication bias assessment is to reduce the risk of false positives or Type I errors. Setting a more lenient threshold, such as p < 0.10, increases the likelihood of incorrectly concluding that there is publication bias when it may not actually be present.

When assessing publication bias, researchers often use various statistical tests and graphical methods, such as funnel plots, Egger's test, and Begg's test. These tools help evaluate whether there is a systematic relationship between study precision and effect size, which could indicate the presence of publication bias.

2) Please Revisit the exclusion criteria. I am giving some suggestion. It is not mandatory to include all suggestion. yet your study exclusion criteria is important

Study Design:

Exclude studies that are not primary research articles (e.g., reviews, editorials, commentaries).

Exclude observational studies lacking a clear dose-response relationship or studies not reporting relevant data for a dose-response analysis.

Exclude studies with insufficient methodological detail or poor quality (e.g., high risk of bias, inadequate statistical analysis).

Population:

Exclude studies that focus exclusively on populations other than patients with type 2 diabetes mellitus.

Exclude studies with mixed populations where data on patients with type 2 diabetes cannot be extracted separately.

Exclude studies with comorbid conditions that could independently affect cognitive function (e.g., neurodegenerative diseases other than diabetes-related cognitive impairment).

Intervention/Exposure:

Exclude studies that do not report hypoglycemic events or provide insufficient information on the exposure.

Exclude studies where the dose or severity of hypoglycemic events is not clearly defined.

Exclude studies where the exposure assessment does not align with the objectives of the dose-response meta-analysis.

Outcome:

Exclude studies that do not report cognitive impairment as an outcome.

Exclude studies that use different definitions or measurements of cognitive impairment.

Exclude studies with insufficient data for the dose-response relationship or studies not reporting relevant effect measures.

Publication Characteristics:

Exclude studies published in languages other than those your team can review.

Exclude studies with incomplete or inaccessible data, and those without sufficient details to conduct a dose-response analysis.

Duration and Follow-up:

Exclude studies with a duration that is too short to capture meaningful cognitive changes.

Exclude studies with inadequate follow-up periods or those not reporting relevant follow-up data.

Publication Date:

Consider excluding older studies if there have been significant advancements in the understanding of diabetes-related cognitive impairment over time.

Intervention Heterogeneity:

Exclude studies with significant heterogeneity in interventions or treatment protocols that may impact the ability to conduct a meaningful dose-response analysis.

3) Please give two heading study strength and study limitation and explain in depth.

4) The rational to use Grading the quality of evidence is not clear. Explain in depth. It is suggestion compare GRADE with AMSTAR 2 (A MeaSurement Tool to Assess systematic Reviews), AHRQ (Agency for Healthcare Research and Quality) Methods Guide and ROBIS (Risk of Bias in Systematic Reviews) Tool.

---

## [Author Response · Author response to Decision Letter 2]

14 Dec 2023

Response to journal's requirements

Ans: I appreciate your kind reminder. We have conducted a thorough investigation of each reference once more by your recommendation. To standardize the reference format, all references were downloaded from PubMed and entered into the Endnote program. There is no issue number for references 7, 12, 13, 14, 38, 40, and 44, and there is no digital object identifier number for reference 48 since it is not included in the original journal. Simultaneously, we examined all the references and were happy to see that none of the withdrawn publications had been cited. Once again, I appreciate your thorough review. 

Response to Reviewer 3's Comments

Firstly, we would like to express our sincere thanks for your constructive comments and positive response. The revised version of the manuscript has been significantly modified based on your comments. The revised parts are marked in red font. We hope you are satisfied with our revisions.

1. “There will be four categories for heterogeneity: not important (0%-40%), moderate (30%-60%), substantial (50%-90%) and considerable (75%-90%). The fixed effect model will be applied for combined analysis if the I2 value is less than 50%. A random effect model will be applied when P < 0.1 or I2 > 50%[32, 33]. Subgroup analysis and other techniques will be used to investigate sources of clinical or methodological heterogeneity when I2 > 75%.” - I suggest that considerable is 75-100%. I also suggest including subgroup studies even if there is no detected statistically significant heterogeneity. 

Ans: Regards for your insightful recommendations. About the I2 value classification, while there are currently a variety of versions, following a collaborative discussion among all authors, we consider your recommendations to be more feasible and scientifically sound, so we modify the considerable heterogeneity to 75–100% by your recommendations. We made adjustments for assertions that were not accurate. That is, regardless of whether statistically significant heterogeneity is found, subgroup analysis and other workable analytical techniques will be used to investigate clinical heterogeneity if there is enough literature. Details are shown in lines 192-197.

2. Publication bias – Why is p < 0.10 not used to assume publication bias? It is the most commonly used value. 

Ans: We appreciate your suggestion, but regrettably, we would still like to use P < 0.05 as the standard for determining publication bias for the reasons listed below.

1. After searching across several databases for relevant meta-analyses, we discovered that the most widely used value for assessing publication bias remains 0.05, with only a tiny number of studies using 0.10. 

2. The commonly used threshold for statistical significance in scientific research is P < 0.05, which means that the observed results are unlikely to have occurred by chance alone. However, the use of P < 0.10 to assume publication bias is not a standard practice. The threshold of P < 0.05 is widely accepted in the scientific community and is considered a standard level of significance. 

3. The rationale behind using a stricter threshold for publication bias assessment is to reduce the risk of false positives or Type I errors. Setting a more lenient threshold, such as p < 0.10, increases the likelihood of incorrectly concluding that there is publication bias when it may not be present. 

 

Response to Reviewer 4's Comments

Firstly, we would like to express our sincere thanks for your constructive comments and positive response. The revised version of the manuscript has been significantly modified based on your comments. The revised parts are marked in red font. We hope you are satisfied with our revisions.

1. Line 85: I propose to rephrase the objective. I suggest the following: "This manuscript aims to establish a methodological structure (protocol) to explore the cumulative effect of frequency of hypoglycemic events on cognitive function and further explore the dose-response relationship through systematic review and meta-analysis, which aims to update the association between hypoglycemia and cognitive impairment by including the latest relevant studies.".

Ans: We appreciate your patient advice, and we have adapted the research purpose in light of your recommendations, which improves the overall quality and scientific appearance of our program. For information, see lines 85–89.

2. Could the authors please cite the MOOSE statement?

Ans: We appreciate you reminding us and apologize for this error. As per your recommendation, we have now included the pertinent references of the MOOSE statement. For information, see lines 96–97 and 393–396.

Response to Editor's Comments

1. As an editor, suggest it to keep statistical significance in scientific research is p < 0.05. I have given my details below. Moreover, Please explain in detail the Publication bias. The current has no justification and explanation.

Ans: Dear editor, thank you for your advice. After consultation with all the authors, we decided to adopt your suggestion and retain a P value of less than 0.05 as the measure of publication bias. For the following reasons:

(1) Within the scientific world, the threshold of P<0.05 is commonly acknowledged as the conventional level of significance. We examined meta-analyses from several databases, and it was found that the majority of writers continued to use P< 0.05 as the standard for evaluating publication bias.

(2) The reviewers recommend a looser criterion of P < 0.10, which will boost the sensitivity of the detection of publication bias. However, it may also lead to an incorrect conclusion that publication bias may be present in the findings when it might not be. As a result, we eventually chose to keep P < 0.05 as our publishing bias metric.

(3) Furthermore, we have included the potential for publication bias in our study protocol in the text. There was an explanation of pertinent potential publishing bias in detail. Refer to lines 214–222.

2. Please Revisit the exclusion criteria. I am giving some suggestion. It is not mandatory to include all suggestion. yet your study exclusion criteria is important.

Ans: Thank you for your valuable advice, which is very important to us. We have appropriately supplemented the exclusion criteria according to your suggestion, which makes our study protocol more rigorous and complete. See lines 137-149 for details. However, we did not use all of these recommendations as exclusion criteria for the following reasons:

(1) The inclusion criteria refer to the basic conditions for enrollment, while the exclusion criteria should be other special conditions that do not meet the requirements of the trial based on the inclusion criteria, rather than the opposite of the inclusion criteria. Therefore, suggestions about the opposite of the inclusion criteria, such as “studies that focus exclusively on populations other than patients with type 2 diabetes mellitus”, “studies with mixed populations where data on patients with type 2 diabetes cannot be extracted separately”, “studies that do not report hypoglycemic events”, “studies that do not report cognitive impairment as an outcome”, “studies that use different definitions or measurements of cognitive impairment”, “studies that are not primary research articles (e.g., reviews, editorials, commentaries)”, and “studies not reporting relevant effect measures” will not be included in the exclusion criteria, because these studies would not have been considered for inclusion in the first place.

(2) Furthermore, as stated in the inclusion criteria, original studies can be included as long as they provide pertinent data for the dose-response relationship, and the dose-response relationship was established by a meta-analysis synthesizing data from multiple original studies. Therefore, "observational studies lacking a clear dose-response relationship" will not be considered an exclusion criterion. All initial research doesn't need to have been done with dose-response relationship calculations.

(3) The results of the literature pre-search indicate that there hasn't been any notable advancement in our understanding of diabetes-related cognitive impairment, and the inclusion criteria specify the methodologies used to measure cognitive impairment. As there is nearly no earlier research remaining after rigorous screening in accordance with the inclusion criteria, it is not required to consider omitting earlier research at this time.

3. Please give two heading study strength and study limitation and explain in depth.

Ans: We appreciate your proposal, and as stated in lines 290 to 305 of the text, we added both the strengths and limitations of this study protocol.

4. The rationale to use Grading the quality of evidence is not clear. Explain in depth. It is suggestion compare GRADE with AMSTAR 2 (A MeaSurement Tool to Assess systematic Reviews), AHRQ (Agency for Healthcare Research and Quality) Methods Guide and ROBIS (Risk of Bias in Systematic Reviews) Tool.

Ans: We appreciate your constructive guidance. We examined four different tools (GRADE, AMSTAR 2, AHRQ, and ROBIS) based on your recommendations. In the end, we chose to evaluate the quality of the evidence using two tools: GRADE and ROBIS (see lines 36–37 and 228–236). The following are the reasons.

(1) AMSTAR 2 applies to systematic reviews of randomized controlled trials, non-randomized intervention studies, or both. This study protocol will include observational studies to explore the relationship between exposure factors and outcome measures and does not involve intervention studies, so AMSTAR 2 is not suitable for this study protocol.

(2) AHRQ Methods Guide recommends the criteria for quality assessment of observational studies, which assesses the risk of bias from five domains: selection bias, performance bias, follow-up bias, measurement bias, and reporting bias. It can assess the risk of bias in different types of original studies included in the same systematic review but does not give recommendations on how to determine the overall risk of bias in the review. As for the original study, the NOS scale, which is more commonly used and convenient, is intended to be used in this study. The NOS scale is a good combination of the reality of case-control studies and cohort studies, which is more in line with this study protocol. Therefore, the AHRQ Methods Guide will not be used in this study.

(3) Unlike the AMSTAR 2 tool, which concentrates on evaluating the methodological quality of systematic reviews, ROBIS is committed to evaluating the risk of bias in the production of systematic reviews. ROBIS is suitable for our research protocol because it may be used for systematic reviews of many study types, such as prognostic, etiologic, interventional, and diagnostic test reviews.

(4) The GRADE criteria explicitly define the quality of evidence and the strength of recommendations, completely abandoning the practice of grading evidence based on the type of study design and instead considering the type of study design, methodological quality, consistency of results, and directness of evidence.

(5) The methodological quality of systematic reviews reflects only part of the risk of bias, so it seems more reasonable for reviewers to evaluate the quality of systematic reviews using different tools at the same time, complementing each other. For the final evaluation of the quality of the evidence, we therefore chose to employ the GRADE criteria and ROBIS, taking into account the specific circumstances of the study protocol.

---

## [Editor Report · Decision Letter 3]

18 Dec 2023

Association of hypoglycemic events with cognitive impairment in patients with type 2 diabetes mellitus: Protocol for a dose-response meta-analysis

PONE-D-23-23141R3

Dear Dr. Yuan,

We’re pleased to inform you that your manuscript has been judged scientifically suitable for publication and will be formally accepted for publication once it meets all outstanding technical requirements.

Kind regards,

Muhammad Shahzad Aslam, Ph.D.,M.Phil., Pharm-D

Academic Editor

PLOS ONE
---

## [Editor Report · Acceptance letter]

25 Jan 2024

PONE-D-23-23141R3 

PLOS ONE

Dear Dr. Yuan, 

I'm pleased to inform you that your manuscript has been deemed suitable for publication in PLOS ONE. Congratulations! Your manuscript is now being handed over to our production team.

Kind regards, 

on behalf of

Dr. Muhammad Shahzad Aslam 

Academic Editor

PLOS ONE